# The Prevalence and Risk Factors Associated with the Presence of Antibiotic Residues in Milk from Peri-Urban Dairy Cattle Farms in Kathmandu, Nepal

**DOI:** 10.3390/antibiotics14010098

**Published:** 2025-01-16

**Authors:** Erda E. Rame Hau, Minu Sharma, Bal K. Sharma Khanal, Peter D. Sly, Deirdre Mikkelsen, Nicholas Clark, Ricardo J. Soares Magalhães

**Affiliations:** 1Queensland Alliance for One Health Sciences, School of Veterinary Science, The University of Queensland, Gatton, QLD 4343, Australia; 2National Zoonoses and Food Hygiene Research Centre (NZFHRC), Kathmandu, Nepal; 3Ministry of Agriculture and Livestock Development, Kathmandu, Nepal; 4Children’s Health and Environment Program, Children Health Research Centre, The University of Queensland, Brisbane, QLD 4000, Australia; 5School of Agriculture and Food Sustainability, The University of Queensland, Brisbane, QLD 4072, Australia

**Keywords:** antibiotic residues, antibiotic stewardship, dairy cattle, peri-urban Kathmandu, risk factors

## Abstract

**Background/Objectives:** The presence of antibiotic residues (ARs) in animal products such as milk can be an important driver of antimicrobial resistance in commensal and pathogenic bacteria. Previous studies on ARs in Nepal have demonstrated the presence of ARs in milk samples but without further characterization of the samples for risk factor analysis. This study aimed to quantify the prevalence and risk factors for the presence of ARs in 140 peri-urban dairy farms in Kathmandu, Nepal, included in a cross-sectional survey in 2019 to estimate farm-level AR prevalence. **Results:** Our results reveal the presence of ARs of sulfamethazine (61%), sulfamethoxazole (53%), ciprofloxacin (46%), and enrofloxacin (42%). Furthermore, of those samples positive for sulfamethazine, sulfamethoxazole, and ciprofloxacin, 81%, 42%, and 42%, respectively, exceeded the maximum residue limit (MRL). While samples taken from farms where staff administered antibiotics were less likely to have single drug residues and multidrug residues (two, three, and four drugs), farms with more workers were more likely to have single residues. Moreover, samples from farms with a higher number of calves and milking cows were more likely to contain single and multiple residues exceeding the MRL, while milk from farms with higher numbers of dry cows and farmers reported by a visiting chemist were less likely to have multidrug residues exceeding the MRL. **Methods:** High-performance liquid chromatography was conducted on bulk milk samples from farms for an AR analysis, revealing positive results. Additionally, a structured questionnaire and direct farmer interviews were used to collect data on farm biosecurity and farming practices, animal health and hygiene, antibiotic usage (AMU), and attitudes and practices towards antibiotic stewardship. **Conclusions:** Ultimately, this study provides evidence on the role of modifiable ARs risk factors in the peri-urban milk industry of Kathmandu, which can serve as a foundation for developing improved antimicrobial stewardship guidelines and designing intervention measures to reduce the public health risk posed by ARs in milk sold in Kathmandu, Nepal.

## 1. Introduction

Antibiotics are commonly used in farmed animals for metaphylaxis, prophylaxis, and growth promotion [1]. They can be administered directly to animals or through feed and water to prevent disease, reduce disease severity, or improve weight gain and feed efficiency [2]. However, misuse of antibiotics in livestock can lead to the accumulation of antibiotic residues (ARs) in animal tissues or products, contributing to antimicrobial resistance (AMR) development in both pathogenic and commensal bacteria [3]. The presence of antibiotic residues, along with other contaminants such as aflatoxins, in milk and dairy products is a significant global health concern [4,5].

ARs are defined as “pharmacologically active substances (whether active principles, recipients, or degradation products), and their metabolites which remain in foodstuffs obtained from animals to which the veterinary medicinal products have been administered” [6]. Despite being metabolized and excreted in urine and feces, these substances can still be detected in animal products and byproducts, such as meat, milk, and eggs [7].

Several factors can contribute to ARs in animal products. These include failure to adhere to withdrawal periods, off-label dosages, contamination of feed with excreta from treated animals, and the use of unlicensed antibiotics [8]. Tetracyclines, sulphonamides, and quinolones are among the most commonly used veterinary drugs in dairy production, with the maximum residue levels (MRLs) for these antibiotic classes being strictly regulated in developed nations by the Codex Alimentarius Commission [9,10,11].

The presence of ARs in food of animal origin can have serious health consequences. Certain drugs may cause direct toxic reactions or hypersensitivity responses [12], such as β-lactam antibiotics (e.g., penicillin and cephalosporin groups), which can induce skin rashes, dermatitis, gastrointestinal issues, and anaphylaxis, even at low doses [13]. Additionally, exposure to low levels of antibiotics can alter intestinal microbiota, potentially leading to the development of resistant pathogens, complicating antibiotic therapy [12].

In the dairy sector, antibiotics are crucial for treating and preventing mastitis, arthritis, respiratory diseases, gastrointestinal infections, and other bacterial infections. In Nepal, the dairy sector is predominantly composed of small-scale production units that play a crucial role in socio-economic development, contributing to 9% of Nepal’s Gross Domestic Product (GDP) and 33% of the agriculture GDP (AGDP), as well as providing employment to two-thirds of the population [14,15]. The majority of the 3.8 million farming households in the country produce milk mainly for home consumption; only about 14% (500,000) of them are both producers and sellers. Only 25% of the total milk produced in Nepal comes into the formal sector, where it is then processed into products [14,16]. Due to the higher demand for milk and milk products in urban areas compared to rural regions, dairies are situated in semi-rural peri-urban places, and the milk is then transported to urban areas for marketing [15]. These peri-urban areas were selected to be included in our study in Nepal because they are the primary producers of Nepalese milk distributed to and consumed by residents of Kathmandu.

Despite the importance of the dairy sector in Nepal, it faces challenges, including inadequate animal feeding, health care, and antibiotic monitoring [17], with the country lacking comprehensive regulations and documentation for antimicrobial stewardship. Research has shown widespread ARs in Nepali milk, with prevalence rates ranging from 5% to 81%, often exceeding the MRLs for certain antibiotics [18,19,20,21]. These results emphasize the importance of comprehensive monitoring and surveillance systems to address potential health risks associated with antibiotic misuse in animal husbandry.

While previous studies on antibiotic residues in Nepali milk have demonstrated that ARs can be found in milk for human consumption and, in some occasions, exceeding MRLs, these studies lacked a broader analysis of associated risk factors [19]. There is a need for larger-scale studies in highly productive regions to not only assess ARs' prevalence in Nepali milk but also identify contributing risk factors. Linking data on the presence and level of ARs in milk with farm-level biosecurity risk factors will allow for the identification of modifiable interventions and antibiotic stewardship strategies that include prescription guidelines and future surveillance and monitoring systems by relevant authorities to address ARs in milk [21,22,23].

This study aimed to determine the farm-level stewardship and contextual socio-demographic factors related to the presence and concentrations of ARs (measured by MRL exceedance) in milk produced in peri-urban districts of Nepal’s capital city of Kathmandu. We hypothesized that i) ARs would be detected in Nepali milk sampled from peri-urban farms and that ii) these ARs concentrations would be below MRL values.

## 2. Results

### 2.1. Dataset for Analysis

A total of 140 enrolled farms were located in seven districts—Balaju, Budhanilkantha, Chandragiri, Godawari, Kirtipur, Ramkot, and Sundarijal—recognized as main milk producers in peri-urban Kathmandu were included in the study (Figure 1). The enrolled farms had an average of 5.24 animals each, with farm sizes ranging from 1 to 11 animals. Most of these animals were milking cows, averaging 3.5 per farm. Farms also reported an average of one calf each, with the number of calves ranging from none to three. Additionally, the majority of farms (79/140) did not have any dry cows, while 33 farms had one dry cow, and 22 farms had two dry cows.

On average, 2.7 people lived on each farm, with women making up the majority, averaging 1.5 women per farm. Each farm employed an average of 5.4 workers, with an equal number of male and female workers. Women were primarily responsible for tasks such as milking (99/140 farms), feeding (93 farms), cleaning (111 farms), and caring for calves (134 farms). In contrast, men were mainly responsible for administering treatments (105 farms) and delivering colostrum (132 farms).

Most farmers (130/140 farms) reported visiting chemists. Among these farms, 93 had chemists located within 1–5 km, while 39 reported chemists at distances of >5 km. Additionally, the majority of farms (124/140) were located >5 km from government veterinary hospitals, with a significant number of farmers (105/140) never visiting these hospitals. Conversely, most farmers (94/140) reported that private veterinary hospitals were within 1–5 km, and almost all farmers (136/140) had visited private veterinary services, such as veterinarians or paravets.

Nearly all farmers (138/140) did not keep records of sick animals or the treatments provided. Additionally, most farmers (132/140) reported that local veterinary authorities or cooperatives did not conduct regular checkups on their animals. While half of the farmers (69/140) reported vaccinating their animals, the vast majority (137/140) did not keep records of the drugs used or administered.

Almost all farmers (132/140) were knowledgeable about antibiotics, and most (135/140) understood that antibiotics should be used to treat sick animals. The majority of farmers (117/140) indicated that antibiotic treatments were primarily administered by contracting company administrators, while some (84/140) also reported administration by farm staff. Over half of the farms (83/140) did not withhold milk after antibiotic use, while the remainder did.

### 2.2. Prevalence of Antibiotic Residues

Out of the 140 milk samples, 60% (84 samples) tested positive for at least one AR (Figure 2). The highest prevalence was found in Budhanilkanta, where 90% (9/10 samples) tested positive for ARs. This was followed by Kirtipur, with a prevalence of 71.4% (15/21 samples), and Ramkot, with 62.4% (5/8 samples). In Sundarijal, 59.3% (32/54 samples) tested positive, while in Balaju, 57.7% (15/26 samples) showed ARs. Only two districts, Chandragiri and Godawari, had an AR prevalence below 50%, with 45.5% (5/11 samples) and 30% (3/10 samples), respectively.

Our results show that sulfonamides were detected in more than half of the samples analyzed. Sulfamethazine was the most prevalent antibiotic residue, found in 65.71% (92/140) of the samples, followed by sulfamethoxazole, which was present in 52.86% (74/140) of the samples (Figure 3). The fluoroquinolone group was found in just under half of the samples, with ciprofloxacin being detected in 46.43% (65/140) and enrofloxacin in 42.14% (59/140). Additionally, sulfamethazine was the most common drug residue exceeding its MRL at 81.16%, followed by sulfamethoxazole (41.89%), ciprofloxacin (41.54%), and enrofloxacin (13.56%).

A spatial analysis revealed that sulfamethazine residues were the most predominant across most locations, found in more than half of the samples, except in Chandragiri, where only about 10% of samples had sulfamethazine (Figure 4). In Chandragiri, ciprofloxacin residues were the most prevalent, detected in 27.3% of samples. In Sundarijal, 32.86% of samples contained sulfamethazine residues, and 20% contained sulfamethoxazole residues, while 31.43% had no detectable ARs. A similar pattern of AR prevalence was also observed in Balaju and Kirtipur.

Regarding drug residues that exceeded the MRL (Figure 5), in Budhanilkanta, all ciprofloxacin residues exceeded the MRL, whereas in Godawari, all residues were below the MRL (Figure 5A). Enrofloxacin residues showed a low percentage of MRL exceedance across four locations (Figure 5B). Sulfamethazine, being the most prevalent residue, had a higher percentage of samples exceeding the MRL compared to residues with a lower prevalence in most locations, except for Chandragiri (Figure 5C). In Chandragiri, none of the sulfamethoxazole residues exceeded the MRL, while in other locations, about a quarter of the samples tested exceeded the MRL (Figure 5D).

### 2.3. The Risk Factors Associated with the Presence of ARs and MRL Exceedance

Our final multinomial model revealed that samples from farms where antibiotics were administered by farm staff had lower odds of containing single drug residues (*p* = 0.01) and multidrug residues. Specifically, the odds of having residues of two, three, and four drugs were significantly lower (OR: 0.14 (0.03–0.61) and *p* = 0.01; OR: 0.25 (0.06–0.92) and *p* = 0.038; OR: 0.19 (0.05–0.71) and *p* = 0.014; and OR: 0.18 (0.04–0.79) and *p* = 0.023, respectively) (Table 1). Additionally, milk from farms with a higher number of workers had increased odds of containing a single drug residue, with an odds ratio of 1.61 (1.1–2.33) (*p* = 0.013).

The final multivariable model indicated that milk from farms with a higher number of calves had increased odds of containing a single residue exceeding the MRL [OR: 2.41 (1.09–5.36), *p* = 0.03]. Conversely, milk from farms with more milking cows had higher odds of containing multidrug residues exceeding the MRL [OR: 1.56 (1.04–2.36), *p* = 0.03]. On the other hand, milk from farms with a greater number of dry cows, and those where farmers visited chemists, had lower odds of multidrug residues exceeding the MRL [OR: 0.32 (0.15–0.69), *p* = 0.004; OR: 0.1 (0.01–0.99), *p* = 0.049, respectively] (Table 2).

The results show that milk from farms where feed supplements were used to enhance production had lower odds of containing enrofloxacin residues exceeding the MRL [OR: 0.009 (95% CI: 0.0002–0.343), *p* = 0.011] but had higher odds of containing sulfamethoxazole residues exceeding the MRL [OR: 15.02 (95% CI: 1.89–120.3), *p* = 0.01] (Table 3). Additionally, milk from farms visited by private veterinary services had lower odds of having enrofloxacin residues exceeding the MRL [OR: 0.009 (95% CI: 0.0001–0.576), *p* = 0.026].

Milk from farms with female staff responsible for milking and those with more frequent visits from a private veterinary service had lower odds of ciprofloxacin residues exceeding the MRL [OR: 0.107 (95% CI: 0.01–0.79), *p* = 0.029; OR: 0.59 (95% CI: 0.37–0.94), *p* = 0.027, respectively]. Conversely, farms that sold milk twice a day had higher odds of ciprofloxacin residues exceeding the MRL compared to those that sold milk once a day [OR: 41.67 (95% CI: 1.2–1450.98), *p* = 0.039]. Samples from farms with farmers who had high school education had lower odds of sulfamethazine residues exceeding the MRL compared to those with farmers who had no education [OR: 0.22 (95% CI: 0.06–0.8), *p* = 0.022].

Milk from farms with more female workers had higher odds of sulfamethoxazole residues exceeding the MRL [OR: 2.11 (95% CI: 1.1–4.05), *p* = 0.024]. However, milk from farms where the last time an antibiotic was used was between 6 months and 1 year ago had lower odds of sulfamethoxazole residues exceeding the MRL compared to those where antibiotics were used within the past week [OR: 0.03 (95% CI: 0.003–0.34), *p* = 0.004].

Interestingly, milk from farms where farmers did not know that higher doses of these medicines do not cure disease had lower odds of sulfamethoxazole residues exceeding the MRL compared to those who disagreed with this statement [OR: 0.12 (95% CI: 0.02–0.61), *p* = 0.01]. Lastly, farms where farmers agreed to routinely withhold milk from treated animals were less likely to have sulfamethoxazole residues exceeding the MRL compared to those who disagreed [OR: 0.23 (95% CI: 0.04–1.19), *p* = 0.07].

## 3. Discussion

The dairy sector in Nepal plays a crucial role in the national economy, contributing one-third of the agricultural GDP and 4% of total exports [14]. Previous studies have reported significant levels of ARs in milk throughout the country, highlighting inadequate stewardship practices regarding antibiotic use at the farm level [19]. However, a recent study reported a 50% reduction in the availability of veterinary antibiotics in Nepal from 2018 to 2020, dropping from 91,088 kg to 45,671 kg [24].

Despite the recent reduction in antibiotic availability, our study reveals that ARs remain a significant issue in peri-urban dairy farms supplying Nepal’s capital city. Approximately 60% of milk samples tested positive for at least one AR, and more than half of these samples exceeded their respective MRLs. This suggests that farm-level practices related to antibiotic use continue to drive AR presence. Compared to studies in other regions, a study examining antibiotic residues in milk from three popular Kenyan milk vending machines found that 24% of samples from both the vending machines and street vendors tested presumptively positive for at least one antibiotic [25]. A study investigating the prevalence of antibiotic residues in pasteurized and unpasteurized milk, as well as antibiotic-resistant bacteria in unpasteurized milk sold in Kibera, an informal settlement in Nairobi, found β-lactam and tetracycline residues in 7.4% and 3.2% of all milk samples, respectively. Residues were significantly more prevalent in unpasteurized milk (23.8%) compared to pasteurized milk (6.8%) [26]. A survey on antibiotic residues in raw milk samples from six sites within the dairy pool of Niamey, Niger, revealed that out of a total of 192 samples, 19 (9.9%) tested positive, including 10 samples from collection centers and 9 from farms [27].

Previous studies in Nepal did not quantify the risk factors associated with AR presence and MRL exceedance. Our study addresses this knowledge gap by quantifying AR presence and associated risk factors in the peri-urban areas of Kathmandu, which are key milk production regions in the country.

Sulfonamides and quinolones are commonly used veterinary antibiotics in the dairy industry. Sulfonamides, in particular, are used as feed additives due to their growth-promoting effects with prolonged ingestion [28]. They have also been associated with a reduction in respiratory diseases among food-producing animals [29,30]. Sulfamethazine, a frequently used sulfonamide, is used on dairy farms to treat mastitis, respiratory infections, and gastrointestinal tract infections. It works by inhibiting the synthesis of dihydrofolic acid, making it effective against both acute Gram-negative and Gram-positive bacterial infections [31,32]. However, its effectiveness decreases in the chronic stage of infection, especially when a high level of exudate is present [33]. Given its widespread use, sulfamethazine carries a higher risk of inappropriate and excessive use, leading to elevated residue levels in milk. Our results show that sulfamethazine is the most prevalent antibiotic residue, detected in 65% of positive samples, and has the highest proportion of samples exceeding the MRL, with 81% of positive samples surpassing the limit.

Our results suggest that farmer education is a significant predictor of sulfamethazine residues exceeding the MRL. Farms with farmers who had high school education were less likely to have sulfamethazine residues above the MRL. This finding is consistent with previous studies that identified a strong relationship between farmers’ educational levels and their knowledge, attitudes, and practices related to antimicrobial use and resistance [34,35]. Farmers with higher education levels are generally more aware and have a better understanding of proper antibiotic use and the importance of adhering to withdrawal periods [36].

Another important sulfonamide used in dairy cattle treatment is sulfamethoxazole, which showed high levels of antibiotic residues in our study, with 52% of samples testing positive and 42% of those exceeding the MRL. Our analysis indicates that the likelihood of sulfamethoxazole exceeding the MRL was higher on farms that reported adding feed supplements. This finding may be partly explained by the use of sulfamethoxazole as a feed supplement and the lack of proper antibiotic stewardship/management in its use, a practice that is common among many farmers [29].

Adding antibiotics to animal feed is known to improve feed efficiency, increase growth rates, and reduce diseases in animals [28]. However, using antibiotics as feed additives and growth promoters poses significant risks, including contributing to the global antimicrobial resistance crisis. This issue has sparked a controversial global debate over whether to ban such practices. While some countries have banned the use of antibiotics as growth promoters, proponents argue that the benefits of antibiotic use for growth promotion outweigh the risks [37,38]. In many developing countries, the use of antibiotics as growth promoters persists due to unregulated sales and easy access to antibiotics [29].

Our study also found that sulfamethoxazole residues exceeding the MRL in milk samples were more likely on farms with a higher number of female workers. In contrast, these residues were less likely in samples from farms where farmers lacked knowledge about drug effectiveness and from farms that practiced milk withholding after antibiotic treatment. According to the Nepalese Program on Commercial Agriculture for Smallholders and Agribusiness (CASA), female workers often have limited knowledge about antibiotic use and management in dairy farms [14]. Research indicates that farmers’ knowledge and perceptions of antibiotic use and withdrawal periods are critical in reducing the risk of AR MRL exceedance in milk [4,39]. Previous studies have shown that the presence of ARs in dairy products is linked to farmers’ non-compliance with withdrawal periods after treating animals with antibiotics [40].

Enrofloxacin and ciprofloxacin are effective fluoroquinolones used to treat gastrointestinal and respiratory infections, especially in organisms resistant to other common veterinary antibiotics [41]. In dairy cows, fluoroquinolones are administered intramammarily to treat mastitis and are also used in dry cow therapy to prevent mastitis [42,43]. Our findings show that milk samples from farms where women were responsible for milking, as well as from farms with more frequent veterinarian visits, were less likely to have ciprofloxacin residues exceeding the MRL. According to the CASA Nepal dairy sector report, women are the primary caretakers in Nepal’s dairy farms, with around 70% of rural women engaged in dairy farming. Their daily tasks include collecting fodder, milking cows, and handling various aspects of animal husbandry, such as feeding, managing sheds, health care, and reproductive care. In contrast, men are mainly responsible for transporting milk to collection centers, managing income, and milking [14]. A study in Vietnam found that female producers tend to have better farming practices than their male counterparts [44].

A higher frequency of veterinarian visits likely contributes to more accurate disease diagnoses, appropriate antibiotic treatments, and strict adherence to withdrawal periods, which reduces the risk of residues compared to self-treatment by farmers. In low- and middle-income countries, including Nepal, where antibiotic regulation is often weak, disease diagnoses and treatments frequently rely on advice from farm owners or other farmers. Additionally, easy access to antibiotics leads to their overuse and misuse on farms [40].

Our study also found that milk samples from farms selling milk twice daily were more likely to have ciprofloxacin residues exceeding the MRL compared to those selling milk once daily. The increased frequency of sales might be associated with higher levels of ARs exceeding MRLs, potentially due to non-compliance with withdrawal periods. Previous research has shown that farmers are often reluctant to adhere to milk withdrawal periods because of the financial losses associated with discarding milk [34,40].

Enrofloxacin is a bactericidal fluoroquinolone antibiotic primarily effective against Gram-negative aerobes, with limited activity against Gram-positive bacteria, including *Staphylococcus* species. It is widely used in food-producing animals to treat bacterial infections, such as acute clinical mastitis caused by *Escherichia coli* in dairy cattle, and is expected to be commonly used on dairy farms due to its effectiveness [45]. However, because quinolones are classified as critically important antibiotics for human health, their use in animals must be highly restricted and is recommended only in life-threatening situations [46,47].

Our study found a relatively lower level of AR and MRL exceedance for enrofloxacin (14%) compared to other antibiotics tested. Farms visited by veterinary services were identified as a key factor in reducing the likelihood of enrofloxacin MRL exceedance. This finding underscores the importance of veterinary supervision in AMR stewardship, especially in contexts where regulatory enforcement is weak. Veterinary oversight leads to better outcomes through accurate diagnoses, appropriate administration, and adherence to antibiotic withdrawal periods.

In this study, we used ordinal multinomial regression modeling to investigate the factors associated with the presence and levels of multiple antibiotic classes in milk samples. Our results indicate that farmer behaviors related to antibiotic access, farm staff profiles, antibiotic administration practices, and farm demographics were associated with the presence of multiple antibiotic residues in milk and the number of residues exceeding their MRLs.

Antibiotic misuse is common in Nepal, where antibiotics are available over the counter without regulation. This makes it crucial for chemists to educate farmers about proper antibiotic use, potentially impacting residues in milk. Our results show that samples from farms where farmers visited chemists were less likely to have multidrug residues exceeding MRLs. This suggests that farmers who consult chemists may gain better knowledge about antibiotic stewardship, including making appropriate antibiotic choices, administering them correctly, and observing withdrawal periods [48].

Providing farmers with proper guidance on antibiotic treatment could help reduce the risk of ARs exceeding MRLs [24]. Our findings indicate that milk samples from farms where antibiotics were administered by trained farm staff were less likely to contain ARs compared to farms without such practices. However, samples from farms with a higher number of workers were more likely to have ARs in the milk, suggesting that proper antibiotic administration is linked to better knowledge and attitudes towards their use. Recent studies have shown a decrease in the percentage of milk samples with residues compared to previous research, suggesting that increased awareness of antibiotic withdrawal periods and more prudent use of antibiotics are having a positive impact [49]. On the other hand, another study indicated that distributors and retailers may lack knowledge about veterinary drugs, including their potential side effects [50]. This lack of knowledge could contribute to inappropriate antibiotic use by farmers, such as using them as growth promotors or to compensate for poor sanitation and hygiene practices on farms [51].

We also found that farms with a higher number of workers were more likely to have milk samples containing a single AR, possibly due to an increased contamination risk from inadequate hygiene and sanitation practices. Larger workforces may struggle to maintain consistent knowledge, attitudes, and practices across all workers. Previous studies in Nepal have highlighted that poor knowledge of antibiotic use among farm workers is a predictor of inadequate antibiotic stewardship [4,8,52].

In terms of farm demographics, our study identified key predictors of multiple ARs in milk samples, including a higher number of milking cows and calves and a lower number of dry cows. Mastitis, a common reason for antibiotic use in dairy cows, often results in losses due to reduced milk yield and quality. A study reported that antibiotics are frequently administered to milking cows to prevent mastitis, support milk production, and treat clinical and subclinical mastitis, with many farmers not observing the withdrawal period [53]. Poor hygiene and sanitation practices on dairy farms in Nepal increase the risk of mastitis, leading to greater antibiotic usage, especially on farms with a more cows. This, combined with the failure to observe withdrawal periods, increases the risk of multidrug residues exceeding MRLs in the milk produced. Our final multivariable model shows that farms with a higher number of calves were more likely to have milk samples with single antibiotic residues exceeding MRLs. Additionally, farms with more milking cows tend to have more calves, increasing the likelihood of antibiotic usage and resulting residues in milk.

This study’s findings need to be interpreted in light of its limitations. Firstly, our assessment of risk factors relied on data provided by farmers through questionnaires, and we were unable to verify the accuracy and reliability of this information. In Nepal, farmers are not required to record antibiotic usage in animals, making it difficult to track the types, purposes, doses, and durations of antibiotics used. This highlights the need for supportive government regulations to ensure better record-keeping, which is essential for improved antibiotic stewardship and surveillance. Secondly, the study’s results are based on a smaller sample size (8% fewer farms) than initially planned due to logistical constraints accessing and enrolling peri-urban farms. Despite this, the relatively high completeness of information provided by the enrolled farms helps to mitigate this limitation. Finally, this study constitutes a cross section of the relationship between the presence and MRL of ARs in milk and the associated risk factors, but it is likely that this association changed over time due to the dynamic of ABU within the dairy herd. This will only be ascertained with a longitudinal study design.

## 4. Materials and Methods

### 4.1. Study Design

The sampling frame for this study included all peri-urban districts of Kathmandu, which are major milk production areas supplying the capital city. The sampling process was conducted in two stages: the first stage involved selecting the peri-urban districts, and the second stage involved choosing farms within these districts from which milk samples would be collected. This study included all seven districts identified as the primary milk-producing areas in peri-urban Kathmandu: Balaju, Budhanilkantha, Chandragiri, Godawari, Kirtipur, Ramkot, and Sundarijal.

The total number of farms across the seven selected districts was 438, with farm distributions as follows: 57 in Sundarijal, 114 in Balaju, 34 in Ramkot, 93 in Kirtipur, 49 in Chandragiri, 48 in Godawari, and 43 in Budhanilkantha. The numbers of cattle reported in these districts were 246, 224, 166, 308, 153, 257, and 247, respectively (see Table 4). This yields an average of 4 animals per farm.

Using probability proportional sampling to estimate the sample size from the 438 farms (clusters) within an average of 4 animals per farm, the required sample size was calculated to be 154 farms. Consequently, the sampling plan required samples from 20 farms in Sundarijal, 40 in Balaju, 12 in Ramkot, 33 in Kirtipur, 17 each in Chandragiri and Godawari, and 15 in Budhanilkantha.

Due to logistical constraints, the final sample included 140 farms, which is 8% fewer than initially planned. From these farms, farm-level milk samples were collected along with data on farm demographics, farming practices, outreach services, antibiotic use, and farmers’ knowledge and perceptions of antibiotics through a cross-sectional survey involving farm owners.

Pooled milk samples were collected through cross-sectional surveys from 140 dairy farms across seven locations in the peri-urban area of Kathmandu, Nepal. The locations included Sundarijal (54 farms), Budhanilkantha (10 farms), Balaju (26 farms), Ramkot (8 farms), Chandragiri (11 farms), Kirtipur (21 farms), and Godawari (10 farms).

At each farm, milk was collected in sterile falcon tubes (50 mL capacity, Fisher Scientific) individually from each animal and placed into a common storage tank (20 L capacity with a temperature around 5 °C, made from either plastic or aluminum). Samples were then taken from these tanks after milking before the tanks were transported to the city for sale. For transportation, plastic falcon tubes were placed in a cooler box with several ice packs to maintain the temperature. The pooled milk samples were collected from the storage tanks at each of the 140 dairy farms. The tubes were coded and transported to the laboratory under cold conditions, and then the samples for high-performance liquid chromatography (HPLC) were stored at −40 °C.

### 4.2. Determining Antibiotic Residues in Milk Using HPLC Analysis

The HPLC method was used to quantitatively analyze two antibiotics from the fluoroquinolone group, enrofloxacin and ciprofloxacin, and two from the sulfonamide group, sulfamethazine and sulfamethoxazole. Sample preparation followed a modified protocol based on Malisch [54].

For quantification, 15 g of milk was homogenized with potassium phosphate solution (30 mL). Next, acetonitrile (90 mL) was added in two stages, followed by filtration, aliquotation, and centrifugation (2500× *g*, 15 min). The resulting extract (10 g) was mixed with NaCl (4 g) and a butyl methyl ether/hexane mixture (30 mL) and then centrifuged (2500× *g*, 5 min). The organic layer was combined with ethylene glycol (6 mL) and evaporated in a water bath at 60 °C until reduced to a final volume between 2 and 3 mL.

Thereafter, the ethylene glycol extract was purged with hexane (25 mL), acetonitrile (3 mL), and water (2.5 mL) and then mixed and centrifuged (2500× *g*, 30 min). The acetonitrile/water layer was combined with water (3 mL) and NaCl (500 mg), mixed, and extracted twice with ethyl acetate (15 mL). The resulting extract was mixed with ethylene glycol/acetonitrile (600 μL) and evaporated to dryness. The final dry extract was reconstituted in 1 mL water, filtered through a 0.45 µm nylon filter (J-sil scientific Industry, Agra, India), and 10 µL was injected into the HPLC system.

HPLC details for quinolines are as follows: instrument: 1200 Infinity Series, Agilent Technologies (Santa Clara, CA, USA); sample injection volume: 30 µL; mobile phase composition: methanol, acetonitrile, water, and phosphate buffer; flow rate: 0.08 mL/min; column temperature: 37 °C; detector: Photo Diode Array (PDA) set at 280–320 nm; run time: 15 min; additional mobile phase involving 0.05 M orthophosphoric acid (pH 3.4) and acetonitrile (87:13 *v*/*v*) at a flow rate of 1 mL/min; and detector wavelength: 277 nm. Moreover, HPLC details for sulphonamides are as follows: instrument: 1200 Infinity Series, Agilent Technologies; mobile phase composition: methanol, acetonitrile, water, phosphate buffer; flow rate: 0.08 mL/min; column temperature: 37 °C; detector: Photo Diode Array (PDA) set at 280–320 nm; run time: 15 min; elution details: 10 min isocratic elution with a 35:65 ratio of methanol to 40 mm ammonium acetate in water; flow rate: 1 mL/min; and retention time: sulfamethazine (SMZ) eluted at 4.5 min. The limits of detection (LOD) and limits of quantification (LOQ) for the tested compounds are as follows: ciprofloxacin: LOD = 0.41 ppb, LOQ = 1.3 ppb; enrofloxacin: LOD = 0.52 ppb, LOQ = 1.75 ppb; sulphamethazine: LOD = 8.84 ppb, LOQ = 29.45 ppb; and sulphamethoxazole: LOD = 8.82 ppb, LOQ = 29 ppb. The concentration of antibiotic residues was compared with the respective MRL for each antibiotic.

### 4.3. Risk Factor Data

A structured questionnaire and direct interviews were carried out to gather data on farm biosecurity characteristics, animal health and hygiene, antibiotic usage in dairy production, and attitudes toward antibiotic usage. The collected data also covered general farming and antibiotic usage practices (Appendix A).

### 4.4. Statistical Analyses

Data management and descriptive analysis of laboratory and questionnaire data were conducted using Microsoft Excel 365. Mapping of sampled farms and peri-urban areas, indicating the number and prevalence of detected ARs, was performed using ArcGIS 10.8 [55].

Multinomial logistic regression models were developed in STATA 17.0 to quantify the risk factors associated with the number of ARs present [none (reference category), 1 AR, 2 ARs, 3 ARs, and 4 ARs] and the number of ARs exceeding the maximum residue limit (MRL) [none (reference category), 1 MRL, 2 MRL, and 3 MRL]. The outcome variables included the presence of ARs (present/absent) and whether ARs exceeded the MRL (below/above MRL) [56,57]. All models included a random effect for the district to account for multiple farms sampled within each area. Risk factors for each ARs exceeding the MRL were evaluated using univariable and multivariable logistic regression, with the outcome of interest being the presence of enrofloxacin, ciprofloxacin, sulfamethazine, and sulfamethoxazole above the MRL. A univariable analysis was first conducted to identify associations between risk factors and each AR exceeding the MRL. Variables with a *p*-value below 0.25 were selected for inclusion in the final multivariable logistic regression analysis. A manual backward stepwise variable selection process was then performed, during which the impact of removing variables on the coefficients of the remaining variables was assessed to identify potential confounders. A variable was considered a confounder and retained in the final model if its removal changed the coefficient of another variable by more than 25%. Risk factors were deemed significant in this study if they were included in the final multivariable model with a *p*-value of less than 0.05.

## 5. Conclusions

Our findings indicate that ARs in milk in peri-urban areas of Kathmandu remain a significant public health concern linked to farm demographics (particularly gender), farming practices, animal health outreach services, and farmers’ knowledge, attitudes, and practices regarding antibiotic use. This study provides crucial baseline data for future in-depth intervention studies aimed at improving antibiotic stewardship in the Nepalese dairy industry. Future research would benefit from connecting AR data with longitudinal patterns of antibiotic use and profiling both commensal and pathogenic bacterial resistance.

## Figures and Tables

**Figure 1 antibiotics-14-00098-f001:**
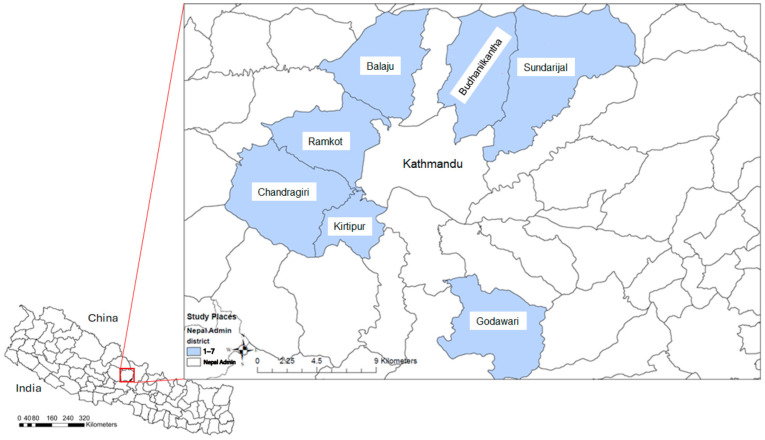
The seven districts, including Balaju, Budhanilkantha, Chandragiri, Godawari, Kirtipur, Ramkot, and Sundarijal, involved in the study in Kathmandu, Nepal.

**Figure 2 antibiotics-14-00098-f002:**
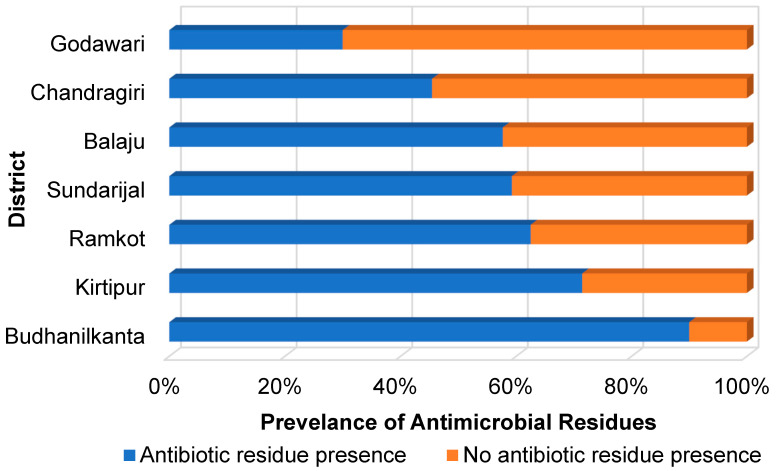
The prevalence of antibiotic residues in the seven peri-urban districts of Kathmandu, Nepal.

**Figure 3 antibiotics-14-00098-f003:**
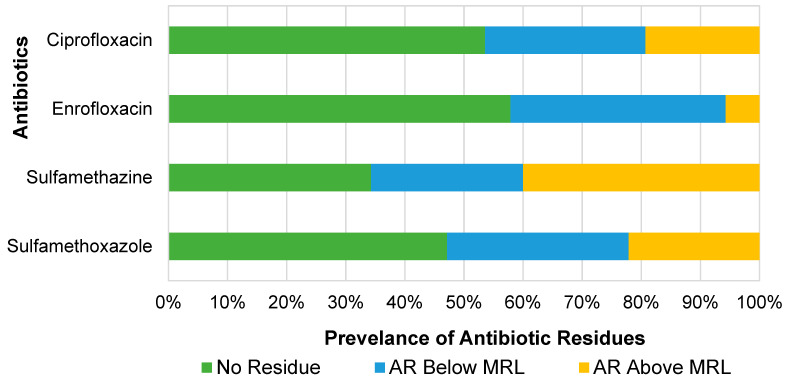
Prevalence of each antibiotic residue (AR) and prevalence of antibiotic residues exceeding maximum residue limit (MRL).

**Figure 4 antibiotics-14-00098-f004:**
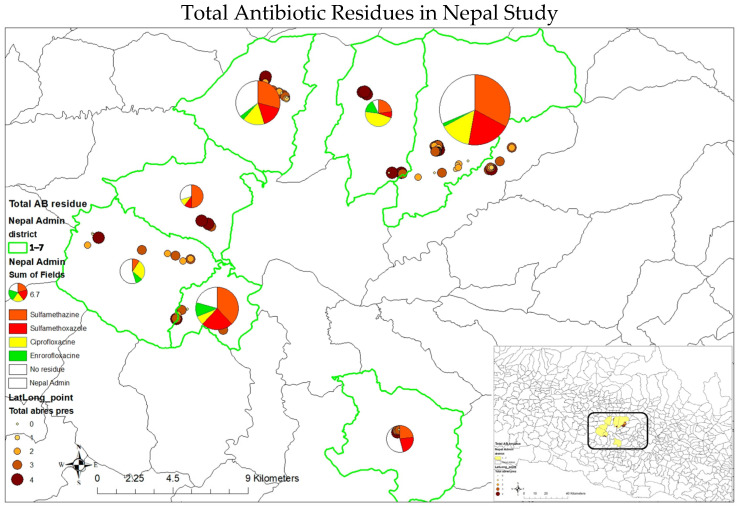
The presence of antibiotic residues in the seven peri-urban districts of Kathmandu, Nepal.

**Figure 5 antibiotics-14-00098-f005:**
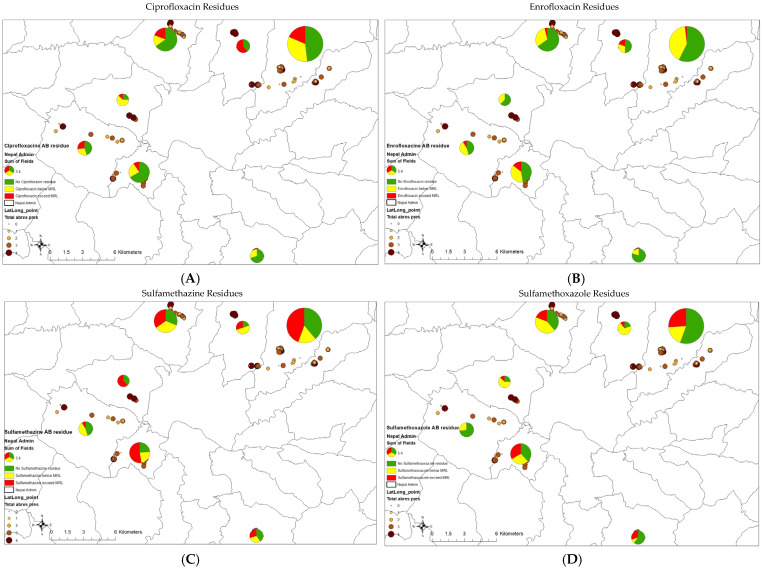
The residue profiles for the antibiotics ciprofloxacin (**A**), enrofloxacin (**B**), sulfamethazine (**C**), and sulfamethoxazole (**D**), indicating the number of antibiotic residues (AB) exceeding the maximum residual limit (MRL) compared to AB presence recorded for the seven peri-urban districts in Kathmandu, Nepal.

**Table 1 antibiotics-14-00098-t001:** The risk factors associated with the number of antibiotic residues (ARs) present in samples.

Variables	Odds Ratio (95% Confidence Interval)	*p*-Value
One AR present (vs. no ARs present)		
Total number of workers (FD)	1.61 (1.1–2.33)	0.013
Women responsible for medicine (FP)	0.25 (0.05–1.28)	0.097
AB administered by farm staff (AMU)	0.14 (0.03–0.61)	0.01
Two ARs present (vs. no ARs present)		
Total number of workers (FD)	1.12 (0.79–1.58)	0.513
Women responsible for medicine (FP)	0.54 (0.16–1.8)	0.325
AB administered by farm staff (AMU)	0.25 (0.06–0.92)	0.038
Three ARs present (vs. no ARs present)		
Total number of workers (FD)	1.43 (0.99–1.95)	0.052
Women responsible for medicine (FP)	0.42 (0.12–1.46)	0.174
AB administered by farm staff (AMU)	0.19 (0.05–0.71)	0.014
Four ARs present (vs. no ARs present)		
Total number of workers (FD)	1.24 (0.84–1.84)	0.262
Women responsible for medicine (FP)	0.5 (0.12–2.07)	0.338
AB administered by farm staff (AMU)	0.18 (0.04–0.79)	0.023

Abbreviations: FD: Farm Demography; FP: Farm Practices; AMU: Antimicrobial Usage.

**Table 2 antibiotics-14-00098-t002:** Risk factors associated with number of antibiotic residues exceeding maximum residue limit (MRL) in samples (no AM residue exceeding MRL (0), 1 AM residue exceeding MRL (1), and 2 or more AM residues exceeding MRL (2)).

Variables	Odds Ratio (95% Confidence Interval)	*p*-Value
One AR exceeding MRL (vs. no ARs exceeding MRL)		
Number of calves (FD)	2.41 (1.09–5.36)	0.03
Number of milking cows (FD)	0.98 (0.7–1.36)	0.903
Number of dry cows (FD)	0.74 (0.42–1.28)	0.282
Women responsible for milking (FP)	0.9 (0.3–2.69)	0.856
Farmers visiting chemists (OS)	1.29 (0.07–22.42)	0.856
Two or more ARs exceeding MRL (vs. no ARS exceeding MRL)		
Number of calves (FD)	1.59 (0.63–4.09)	0.315
Number of milking cows (FD)	1.56 (1.04–2.36)	0.03
Number of dry cows (FD)	0.32 (0.15–0.69)	0.004
Women responsible for milking (FP)	0.37 (0.11–1.29)	0.123
Farmers visiting chemists (OS)	0.1 (0.01–0.99)	0.049

Abbreviations: FD: Farm Demography; FP: Farm Practices; OS: Outreach Services.

**Table 3 antibiotics-14-00098-t003:** Results of multivariable analysis of factors associated with exceedance of MRL for enrofloxacin, ciprofloxacin, sulfamethazine, and sulfamethoxazole.

Variables	Enrofloxacin		Ciprofloxacin		Sulfamethazine		Sulfamethoxazole	
	OR (95% CI)	*p*-Value	OR (95% CI)	*p*-Value	OR (95% CI)	*p*-Value	OR (95% CI)	*p*-Value
Number of calves	4.305 (0.802–23.103)	0.089						
Use vitamin/supplement (Ref: no)	Reference		Reference					
Yes	0.009 (0.0002–0.343)	0.011	2.43 (0.74–8.08)	0.14			15.02 (1.89–120.3)	0.01
Visiting/visited by private veterinary services (Ref: no)	Reference							
Yes	0.009 (0.0001–0.576)	0.026						
Withholding milk (Ref: no)	Reference							
Yes	10.27 (0.9417–112.168)	0.056						
Responsible milking (Ref: men)			Reference					
Women			0.107 (0.01–0.79)	0.029				
Frequency of selling milk daily (Ref: once)			Reference					
Two times			41.67 (1.2–1450.98)	0.039				
Frequency of visits/being visited by private veterinary services			0.59 (0.37–0.94)	0.027				
AB administered by staff (Ref: no)			Reference					
Yes			0.27 (0.06–1.16)	0.079				
Number of milking cows					1.29 (0.97–1.71)	0.078		
Farmers’ education level (illiterate)					Reference			
Primary school					0.49 (0.14–1.69)	0.264		
High school					0.22 (0.06–0.8)	0.022		
Graduate					0.41 (0.06–0.39)	0.339		
Visiting chemist (Ref: no)					Reference			
Yes					0.16 (0.01–1.55)	0.115		
Number of dry cows							1.29 (0.97–1.71)	0.066
Number of female workers							2.11 (1.1–4.05)	0.024
Last time using AB (Ref: within a week)								
Within a month							1.13 (0.14–8.93)	0.899
Three months ago							0.26 (0.03–2.13)	0.214
Six months ago							0.41 (0.06–2.8)	0.364
A year ago							0.03 (0.003–0.34)	0.004
Some of these medicines I use in my dairy farm do not cure diseases even at high doses (Ref: Disagree)								
Agree							0.66 (0.08–5.31)	0.694
Do not know							0.12 (0.02–0.61)	0.01
Routinely withhold milk for at least 7 days from animals which I have given medications (Ref: Disagree)								
Agree							0.23 (0.04–1.19)	0.07
Do not know							0.3 (0.01–3.15)	0.388

OR (95% CI): Odds Ratio (95 Confidence Interval).

**Table 4 antibiotics-14-00098-t004:** Counts of farms, cattle, and sampled farms for each peri-urban district of Kathmandu, Nepal, included in this study.

Study Locations	Farm Count per Location	Total Cattle Count	Sampled Farm Counts
Sundarijal	57	246	20
Balaju	114	224	40
Ramkot	34	166	12
Kirtipur	93	308	33
Chandragiri	49	153	17
Godawari	48	257	17
Budhanilkantha	43	247	15

## Data Availability

The original contributions presented in this study are included in the article/Appendix A, and further inquiries can be directed to the corresponding author.

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
