# Peer review of "The Prevalence and Risk Factors Associated with the Presence of Antibiotic Residues in Milk from Peri-Urban Dairy Cattle Farms in Kathmandu, Nepal"

_antibiotics, 2025, doi:10.3390/antibiotics14010098_

Round 1

Reviewer 1 Report

Comments and Suggestions for Authors

This article is a comprehensive analysis of ARs in milk from peri-urban Kathmandu, highlighting significant public health concerns linked to AMR. The wide geographical coverage, HPLC methodology, and identification of risk factors are strong points of the article. Furthermore, the discussion section effectively connects findings to global health issues and local farming challenges. However, aside from a smaller-than-planned sample size, the study is limited by reliance on self-reported data and a lack of longitudinal analysis. Gender dynamics, though mentioned, require a more profound investigation. Also, more data concerning regulatory recommendations could be added.

All other minor suggestions are in the uploaded file.

Reviewer 2 Report

Comments and Suggestions for Authors

The study addresses an important public health issue: the presence of antibiotic residues (AR) in milk and its role in antimicrobial resistance. This is particularly significant for peri-urban dairy farms, where farming practices can contribute to residue prevalence.

1. Provide more details about the HPLC analysis, such as the detection limits and protocols used, to ensure reproducibility.

2. Compare the prevalence rates of antibiotic residues with those in similar studies from other regions or countries to contextualize the results.

3. Acknowledge potential limitations, such as the cross-sectional design, which may not establish causation.

4. Ensure consistent use of terms such as "antibiotic residues," "antimicrobial usage," and "maximum residue limit (MRL)" throughout the text.

5. I suggest authors to include this recent article to support this research work: doi: https://doi.org/10.1016/j.bios.2022.114035; doi: https://doi.org/10.1002/agt2.402; doi: https://doi.org/10.3390/foods11233922; https://doi.org/10.1016/j.foodcont.2022.108928

Comments on the Quality of English Language

Need to improve

Reviewer 3 Report

Comments and Suggestions for Authors

Journal: Antibiotics

Manuscript ID: antibiotics-3407811

Title: Prevalence and risk factors associated with the presence of antibiotic residues in milk from peri-urban dairy cattle farms of Kathmandu, Nepal

Dear Author,

The manuscript entitled “Prevalence and risk factors associated with the presence of antibiotic residues in milk from peri-urban dairy cattle farms of Kathmandu, Nepal” describes food safety issue about the presence of antibiotic residues in milk. Additionally, further characterization of the samples was performed for risk factor analysis. The manuscript is interesting, however various parts require improvement. Therefore, a minor revision is suggested. Please follow the comments below:

-        Line 23: after “positive” please add dot (.)

-        In the introduction section, please give literature information about the importance of further characterization of the samples for risk factor analysis. The main purpose and novelty of this study is further characterization of samples. Which needs additional literature information in the introduction section.

-        In the introduction section, please give information about peri-urban dairy cattle farms of Kathmandu, Nepal. The reason for selection of this region in this study or the importance of this region regarding antibiotic residues. Please explain with previous studies.

-        Line 102, 119, 264, 275: The term of perimilk is valid or not? In previous studies the use of the term of perimilk cannot been found. I suggest the term of peri-urban milk, or you suggest a more suitable term and use that term.

-        Line 125: please give information storage tank (temperature, volume, and other technical properties)

-        Line 129: please give information about collection tube such as brand, manufacturer, etc.

-        Line 131: please give information about HPLC such as brand, manufacturer, etc.

-        Line 137-146: Please give information about all chemicals (potassium phosphate, acetonitrile etc.) such as brand, manufacturer, concentration etc.

-        Line 139: please give information about centrifugation such as brand, manufacturer, etc. This information is important because centrifugation conditions were given in “rpm”.  Additionally, it is preferable to give it in “g”.

-        Line 132-line 142: please be careful about degree. Which one is true? 0C or oC

-        Line 148: please give information about nylon filter such as brand, manufacturer etc.

-         Line 174: please explain the reason for the selection of variables with a p-value below 0.25. Give some literature information about it.

-        Line 157: Where is Supplementary File 1? I would like to check this file.

-        Line 182: “3.1. Dataset for analysis” please explain the source of these data before following sentence. I suppose it is supplementary file 1?

“The 140 enrolled farms had an average of 5.24 animals each, with farm sizes ranging from 1 to 11 animals.”

-        The results section is clear.

-        The discussion section is clear.

-        Line 365: please revise “researches indicate that ……………………….. (4, 35)”.

-        Line 395: please revise “researches have shown………………………………..”

-        In conclusions section, I think information about the importance of this study (line 470-472) is not necessary.  Please give the main findings about prevalence of antibiotic residues and results with the exceedance of MRL for antibiotics.

-        Please check all references according to spelling rules. For example, in line 502, 506, 508, 518, 520, 524, 533, 535, 546, 562, 570, 573: is the use of uppercase true or not?
